# Working for the Welfare: Support and Supervision Needs of Indigenous Australian Child Protection Practitioners

**Fiona Oates** [1,*]  **and Kaylene Malthouse** [2]

1   College of Arts, Society and Education, James Cook University, Cairns 4870, Australia
2   Private Consultant, Cairns 4870, Australia; nalpajuwi@hotmail.com
*   Correspondence: Fiona.Oates@jcu.edu.au

**Abstract:** Aboriginal and Torres Strait Islander children are disproportionately represented in all parts of the child protection system in Australia. The recruitment of Aboriginal and Torres Strait Islander practitioners into child protection systems to work with Indigenous families at risk underpins the government strategy to reduce this over-representation. However, little is known about the experiences of Indigenous people who undertake child protection work or what their support and supervision needs may be. This research is centered on Indigenous Australian child protection practitioners as experts in their own experiences and as such includes large excerpts of their own narratives throughout. Practitioner narratives were collected via qualitative semi-structured in-depth interviews. Critical theory and decolonising frameworks underpinned the research design. The study found that Indigenous child protection practitioners have a unique place in the families, communities and profession. Many viewed their work in the child protection field as an extension of their Indigeneity. This coupled with the historical experience of state-sanctioned removal of Indigenous children during colonisation and contemporarily, informs the need for child protection workplaces to re-think the support and supervision afforded to Indigenous practitioners.

**Keywords:** Indigenous child protection workers; child protection; Indigenous child protection; staff support and supervision



## 1. Introduction

Child protection work is one of the most complex and demanding areas of social work and human services practice. Central to the work are the practitioners who work to ensure the care and protection needs of vulnerable children are met. In Australia, Aboriginal and Torres Strait Islander children and families are disproportionately represented in the statutory child protection service system. For the period 2018–19, Aboriginal and Torres Strait Islander children were eight times more likely to be in receipt of child protection services (166 per 100,000 compared to 21 respectively) inclusive of investigation and assessment as well as out of home care in Australia (Australian Institute of Health and Welfare 2021).

Disproportionate representation of First Nations children is not unique to Australia. A number of countries where the forced removal of First Nations children occurred as part of colonisation, like Canada, New Zealand and the United States, also have disproportionate numbers of First Nations children receiving child protection services. Despite ongoing system and practice reform including significant financial investment by successive governments in Australia, the numbers of Indigenous children requiring child protection services remain disproportionate with increases predicated over the medium term (Bennett 2013; Briskman 2014; Lewis and Burton 2014). In an attempt by governments to address the disproportionate representation of Indigenous children receiving child protection services, many state-run child protection agencies have adopted strategies to increase recruitment of Indigenous practitioners into the child protection service system, including in Queensland,

where this research took place (Carmody 2013). Social work academic Briskman (2014) argued that employing more Indigenous people in social service structures may generate stronger awareness of Indigenous disadvantage and therefore better equip non-Indigenous practitioners to work more effectively with Indigenous families and communities. However, Briskman (2014) also argued that increasing numbers of Indigenous workers in the child protection service system in an attempt to create the structural change required to address disproportionate representation of Indigenous children and families is untested and subsequently cannot be relied on.

To have a strong contextual understanding of the complexity that exists for many Aboriginal and Torres Strait Islander practitioners working in child protection in Australia, one must first understand the history of the State's intervention into the lives of Indigenous people, in particular, the raising of children. When the British arrived in Australia in 1788, they declared the country 'Terra Nullius', meaning the land was owned by no one and therefore free for ownership to be assumed (Atkinson 2002; Krieken 1999). The declaration of 'Terra Nullius' was the framework from which the colonisers deemed Aboriginal people as flora and fauna and therefore not capable of self-agency. Social evolutionary theory, Darwinism and anthropological scholarship were all used to solidify the colonisers' belief that Aboriginal people were incapable of evolving and assimilating into European society and therefore would eventually die out (Bennett 2013; Muecke 1992; Reynolds 1987; Tatz 1999).

A period of protectionism began in the late 1800s, underpinned by the colonisers' belief that the Aboriginal population would soon naturally expire. During the protectionist period, Aboriginal people were segregated from the settler community, moved from their traditional lands onto missions or settlements and were not afforded any autonomy over their lives (Bennett 2013). Traditional cultural practices like the use of tribal names, participation in Ceremony and speaking language were strictly forbidden (Reynolds 1987). Despite the belief of the colonisers that the Aboriginal population would eventually die out, the Aboriginal population increased, in large part due to relationships, often non-consensual, between settler men and Aboriginal women (Bennett 2013).

The colonisers believed that 'half-caste' children should be protected from their Aboriginality and the best way to do that would be to separate them from their Aboriginal families (Paten and Robinson 2008). Legislation was introduced within the British colonies of Australia which allowed the State to assume legal guardianship of all Aboriginal children, without parental consent or the right of appeal (Bennett 2013). Aboriginal children were systematically removed from their families and communities, and sent to non-Indigenous families by way of adoption, foster care or as 'workers' performing farmhand type work or domestic duties (Atkinson 2002). Others were sent to orphanages or group homes. Children removed from their families as a result of these policies are known in Australia as the 'Stolen Generations' (HREOC 1997). In addition to the trauma resulting from removal from their families and communities, many of the Stolen Generations were subject to ongoing sexual, physical and psychological abuse as well as severe neglect in state-run institutions and foster homes (Atkinson 2002; HREOC 1997).

The social work profession played a central role in the forced removal of Indigenous children from their families and communities. Indigenous Australian social work academic Bennett (2013) describes that at the time social work was an 'instrument of social control' (p. 19) and that social workers were 'participants in the process of dispossession and oppression' of Aboriginal people and communities (p. 20). Many Aboriginal and Torres Strait Islander people and communities have a deep sense of suspicion and distrust of social workers related to the role they played in the state-sanctioned forced removal of Indigenous children (Gilbert 1993; Harms et al. 2011). Contemporarily, this is of particular relevance when considering what role Aboriginal and Torres Strait Islander practitioners in the child protection service system have, what their experiences might be within this context and how best to support them.

This paper is structured in a way that facilitates participants to speak about the rewards and challenges they experience as child protection practitioners within a professional, personal, family and community context and how those experiences inform their support and supervision needs. Throughout this paper, participant practitioner voices are prominently featured, recognising their expertise both in relation to contemporary child protection practice and in their own experience as Indigenous people who undertake child protection work. Discussion framed with a critical lens is woven throughout the paper.

### 1.1. An Acknowledgement

We the authors acknowledge the tens of thousands of years that Aboriginal and Torres Strait Islander peoples raised strong, healthy children in Culture on Country. We acknowledge the hard work and dedication of all the Aboriginal and Torres Strait Islander peoples who have and continue to fight for the restoration of sovereignty, including the right to raise strong healthy children in Culture on Country, stolen as a result of colonisation. We acknowledge the fight is ongoing both in Australia and for First Nation's people internationally.

### 1.2. A Note on Language

As this study was undertaken in Australia, it is important to define some frequently used language and terminology to help create a shared understanding for the readership.

#### 1.2.1. Indigenous

An Indigenous Australian is a person of Aboriginal and/or Torres Strait Islander heritage, who is accepted as Indigenous by the community with which the person associates (Australian Institute of Aboriginal and Torres Strait Islander Studies 2019). Aboriginal and Torres Strait Islander people belong to two separate and unique cultural backgrounds, with distinct languages, customs and belief systems. Aboriginal Australians are the traditional Custodians of mainland Australia. Torres Strait Islander Australians are the traditional Custodians of the Torres Strait Islands. The terms 'Indigenous', 'First Nations' and 'Aboriginal and/or Torres Strait Islander' have been used interchangeably for ease of the reader, not as an indication that the two cultures are homogenous.

#### 1.2.2. Statutory Child Protection

Statutory child protection refers specifically to administering child protection services under the legislation that governs child protection matters within individual states and territories within Australia. Services administered as part of statutory child protection practice in Australia include investigation and assessment of child abuse and/or neglect concerns, monitoring of at risk children in their homes, the removal and placement of children deemed to not have a parent willing or able to tend to their care and protection needs in out of home care, the reunification of children who have been removed back to their family of origin and the long-term case management of children and young people in the long-term care of the State until their 18th birthday. Statutory child protection services in Australia also include adoption.

#### 1.2.3. Non-Statutory Child Protection

Non-statutory child protection in Australia refers to services provided by non-government, not-for-profit organisations to children and families where child protection concerns have been identified by statutory child protection authorities. They are in most cases funded by the government department responsible for statutory child protection in individual states and territories within Australia. The types of programs and services provided by non-statutory child protection organisations include intensive parenting support, domestic and family violence support and behavior change programs, a range of counselling and other therapeutic supports, foster and kinship care assessment and placement support, as well as support to assist young people transitioning from the long-term care of the state. Some of these organisations are

community controlled, meaning they are governed by an all Indigenous management board and funded by the government to provide non-statutory child protection services specifically to Aboriginal and Torres Strait Islander children and families.

### 1.2.4. Black

In this paper, the term 'Black' refers to the participants' description of themselves and other members of the Aboriginal and/or Torres Strait Islander community in Australia.

### 1.2.5. White

In this paper, the term 'White' is used to indicate all people who are neither Aboriginal nor Torres Strait Islander in cultural background, regardless of skin colour or ethnicity.

## 2. Materials and Methods

The findings presented in this paper are a subset of findings from a larger study that explored the experiences of Indigenous child protection workers based in Queensland, Australia (Oates 2018). The larger study relied on the research participants to answer the primary research question: what are the experiences of Indigenous child protection workers? This primary research question reflected the dearth of research related specifically to the way in which Indigenous practitioners experience undertaking child protection work, particularly in an Australian context.

### 2.1. Ethical Approval

This study received ethical approval from James Cook University Human Research Ethics Committee (approval number: H6266).

### 2.2. Research Design

The research outlined in this paper was framed by critical and decolonising theory. Critical and decolonising theory centers those with lived experience as the experts regarding matters pertaining to them, their lives and their communities (Coram 2011; Kowal et al. 2005). Researchers and researchees in a decolonising theoretical research design, are viewed as co-creators of knowledge (Oates 2020) with the acknowledgement that research should tangibly benefit the group who are the focus of the research and that knowledge created should be used as a vehicle for positive change (Jenkins 2015; Mertens 2003; Prior 2007; Wilson and Bird 2005). The Aboriginal and Torres Strait Islander child protection practitioners who participated in this study are centered as the experts in relation to the primary research question with their voices predominant throughout this paper.

### 2.3. Position of the Primary Researcher

As qualitative researchers can never truly separate themselves from the research (Corbin and Strauss 2015; Creswell 2013; Neuman 2014), we will now locate ourselves within it (Absolon and Willett 2005). I, Fiona Oates, am the primary investigator of the research presented in this paper. I am a non-Indigenous female of Anglo-Australian heritage with strong family and professional ties to Far North Queensland, Australia, where I currently live and work. I am a social worker by undergraduate training and have worked in the statutory and non-statutory child protection sector as a practitioner, supervisor and educator. I acknowledge that Aboriginal Australians are the Traditional Custodians of mainland Australia and that Torres Strait Islander Australians are the Traditional Custodians of the Torres Strait Islands. I further acknowledge that I occupy space on land wherever I go within Australia that was never ceded.

The primary motivation for undertaking this study was to collaborate with Indigenous child protection practitioners to build the knowledge and skills of non-Indigenous practitioners and supervisors. Historically, child protective authorities have played a significant role in the dispossession of Aboriginal and Torres Strait Islander people from their culture

and communities. An in-depth understanding of the experiences of Indigenous practitioners will strengthen the ability of non-Indigenous practitioners to collaborate in an informed way. It is also critical that non-Indigenous supervisors of Indigenous practitioners have a deep understanding of the unique supervision and supports that Indigenous practitioners may need to maintain their personal, cultural and professional safety within the workplace.

### 2.4. Position and Role of the Cultural Broker and Mentor

This study was designed to include collaboration with a Cultural Broker and Mentor. Kaylene Malthouse was the Cultural Broker and Mentor assisting myself as the primary researcher and is the co-author of this paper in acknowledgement of the expertise she contributed throughout the research project. Kaylene Malthouse is an Upper Malanbarra Yidinji woman. Her mother, Grace Mary Ambrum, was a Yidinji Nadjan woman from the Atherton Tablelands in Far North Queensland, Australia. X's Great Grandfather's estate commences at Toohey's Creek waterways on the Atherton Tablelands which runs through the Gadgarra Forest into the Mulgrave River in the Goldsborough Valley, taking in the Gillies range southwest of Cairns in Far North Queensland, Australia.

Kaylene Malthouse is a respected member of her community and has an extensive professional background working in the area of education and child protection. Kaylene Malthouse is a dedicated advocate for her People, demonstrated by her work on various boards including 14 years with the North Queensland Land Council, four of which she was Chair. Her work with the North Queensland Land Council included representing the views of her constituents in the Uluru Statement from the Heart process which advocated for Truth Telling, Treaty, a Voice to Parliament and a standalone body representing Indigenous people in the Federal Australian parliament.

The role and purpose of cultural brokers in research varies in the literature, particularly between disciplines. For the purpose of this study, Meyer's (2010) definition most closely reflected the contribution of Kaylene Malthouse in this study. Meyer (2010) argued that the roles of cultural brokers includes 'the identification and localisation of knowledge, the redistribution and dissemination of knowledge, and the rescaling and transformation of this knowledge . . . brokering knowledge thus means far more than simply moving knowledge—it also means transforming knowledge' (p. 120).

Ongoing socio-political conversations between Kaylene Malthouse and myself during the research period, provided me the opportunity to strengthen my understanding of the continued struggle for Indigenous people to be recognised as the traditional Custodians of Australia, a sovereignty that was never ceded. These opportunities to shift from my mainstream Western lens deepened my understanding of the link between recognition of sovereignty and the right to have autonomy over one's agency, and the ability for Indigenous people to raise healthy children in Culture on Country. This is a demonstration of the transformation of knowledge referred to by Meyer (2010) that can be achieved by the inclusion of cultural brokers in the research process.

### 2.5. Participant Voice

The findings section of this paper privileges the voices of participant practitioners, acknowledging their expertise in their own lives and experiences. Positioning Indigenous voices at the fore challenges traditional Eurocentric research practices which routinely excludes Indigenous voices (Muller 2014). Participant practitioners were invited to choose the pseudonym they would be known as in subsequent publications of the research findings. Many chose to be known by the name of a close relative that had special significance in their lives. Indigenous practitioners re-claiming the names of ancestors' past is an acknowledgement of ancestral connection. A strategy used by colonisers to dehumanise Indigenous people and to sever cultural and tribal connections was to remove their birth names, replacing them with European or infantilised names (Atkinson 2002; Bennett 2013). The reclamation of names returns power to Indigenous people who have had to fight for the right to exercise sovereignty over their land and lives. The significance of being

acknowledged by one's birth name to Indigenous Australians in the context of colonising practices designed to destroy connection, cannot be overstated.

### 2.6. Sampling and Participant Consent

The study research design necessitated the participants to self-identify as Australian Aboriginal and/or Torres Strait Islander child protection practitioners. Participants must have worked with children and/or families where child protection concerns were present. Child protection practice could have been undertaken in a statutory or non-statutory practice context. Interested practitioners were provided with a participant information sheet outlining what the study was about, a list of prompt interview questions and a consent form. The provision of prompt questions to interested practitioners was to provide a clear understanding of what kinds of questions would be asked, therefore providing the opportunity for people to give full and informed consent to participate. Equally, it gave practitioners the power not to participate or to limit their participation to topics they were comfortable with. Interested practitioners were informed that they were under no obligation to participate and that they could withdraw their consent to participate at any time.

### 2.7. Introducing the Participants

In total, thirteen child protection practitioners who self-identified as Australian Aboriginal and/or Torres Strait Islander participated in the study. Collectively the participant group held significant practice experience and expertise across both the statutory and non-statutory child protection context with six having supervisory experience. Nine years was the average length of experiences for the three participants who had statutory child protection experience, 11.5 years for the six participants that had non-statutory experience and 10 years for the four participants who had worked in both statutory and non-statutory child protection contexts. The practitioners' qualifications varied. At the time of participation, nine had Bachelor's degree qualifications (social work, community welfare, psychology), and four had certificate or diploma level qualifications or were studying towards attaining a Bachelor's degree level qualification.

### 2.8. Data Collection and Analysis

Practitioner narratives were attained via semi-structured, in-depth qualitative interviews. This method of data collection facilitated an opportunity for participants to convey their expertise through narrative relative to the research question (Brinkmann and Kvale 2015). Affording participants control over their own narrative in the research process shifts the researcher from the position of expert into the role of co-creator of knowledge in partnership with participants (Potts and Brown 2005; Prior 2007). This data collection method is indicative of a decolonising research framework which is important given my non-Indigeneity.

Interviews were audio-recorded with the consent of participants. One participant did not consent to having their interview audio-recorded, although did consent to notes being taken, summarised and presented back to them for approval. Audio-recorded interviews were professionally transcribed and sent to participants for their endorsement. Qualitative data analysis software package NVivo was used to collate and thematically analyse interview data. Qualitative data analysis is about interpreting the data for meaning (Corbin and Strauss 2015; Creswell 2013; Neuman 2014). As such, initial and subsequent codes were assigned to emerging themes. A comprehensive sub-set of themes and sub-themes were deduced from this data analysis process. Supervision and support that recognises the unique experience of Indigenous people working in child protection in an Australian context was a key theme that emerged from the analysis and is the central theme presented in this paper.

## 3. Results/Findings

*3.1. Practitioner Motivations to Work in Child Protection*

Participants were invited to share what drew them to undertake child protection work. Most participants described helping others to be a core value of their culture and subsequently their identity. For many of the participants, their motivation to undertake child protection work was linked to their identity as Indigenous Australians:

> It's in our DNA. (Isabella)

Many participants felt that it was their role to work with their people and communities to improve the outcomes for children and families. The Indigenous participants stated that they viewed themselves as a conduit between the 'white man's' child protection system, their people and their communities. The participants' views were consistent with Zon et al. (2004), who found that Aboriginal child protection workers in the Northern Territory deeply felt a responsibility to change and improve child protection practices for their people and community. Some participants shared their deep desire to work within their communities, to work with their people and to give back for the upbringing that they felt privileged to have had:

> For me, when people ask me, 'How can you do that job? That would be so horrible', I say that I grew up with a mum and dad and I grew up with a strong grandmother and strong cultural beliefs. For me to go out and see family and mob that haven't been given that same opportunity in life is my way of giving back to my people . . . I feel it's my turn to give back to my people and give the best of myself to them. I know that I had it good—I had a good, strong upbringing. A lot of people weren't able to have that because of the life experiences and the history of what they went through. (Missy)

> I knew as a 15 year old girl that social work was what I was going to do . . . because I grew up with family that helped other people, so that's what I did. (Isabella)

The participants consistently linked privilege to being strongly connected to family and culture. Some participants spoke of their motivation to end the disconnection from family and community felt by Indigenous children in state care. Other participants shared that they had experienced disconnection from their own family and community growing up, and felt a strong desire to assist children and young people in the same situation to reconnect to their culture and community:

> . . . it was important for me to give back to society for the good upbringing I had . . . this is where I want to make the difference. I want to be able to make sure the kids go home to family members. (Rosalyn)

> A professional role in relation to providing cultural advice and support across the office. I took on that role because that's who I am . . . I argue with managers that I'm an Aboriginal officer first . . . working for community and being a member of that community. (Sarah)

There were different views among participants relating to how they viewed themselves individually as child protection workers, influenced by the type of child protection work they undertook. The participants who had worked in the non-statutory area of child protection described themselves as having great empathy for the Indigenous clients with whom they worked. Further, they described themselves as 'helping' and 'supporting' their communities, which was a source of pride for them and often for their families:

> Sometimes when working with other Indigenous families, I can really relate to them in a way . . . I can just picture myself and be like, 'Yeah, you know, this could have been me sometimes and I know exactly what you're going through'. (Grace)

However, those who had worked in the statutory child protection area described conflict within themselves. Their values were challenged constantly, and they expressed despair at being unable to make a tangible difference for their people who were 'in the system':

> It does feel like a fail, and I think I do take that a little bit personally sometimes. Like, you always think there's more that I could have done, or that's another—you know, it's like, it's another Black family in the system. (Grace)

### 3.2. Family Views

The motivation to enter the area of child protection was similar for most of the participants, regardless of the type of child protection work they undertook with children and families. When reflecting on how their families felt about them undertaking child protection work, the participants described a range of responses, ranging from pride to concern:

> My mum was happy ... so proud. My dad actually sat me down and said, 'Are you going to be right with this? I think that you need to really sit down and understand what it is that you're getting into'. (David)

> My mum ... she was going, 'You don't want to go there', and I'm like 'I know, but I think this will be a good experience', so—and it was in the sense that I learnt so much about how the system works, my own values and what I will and will not compromise on. (Emma)

> They think I'm courageous ... they all supported me strongly, understood why I did the job that I do. They didn't like sometimes the long hours that I'd spend at work or the way that it would affect me. (Rosalyn)

Indigenous Australian researchers Green et al. (2013) examined the experiences of Aboriginal and Torres Strait Islander people undertaking a Bachelor of Social Work degree and found that choosing to enter the profession of social work can draw harsh criticism from Aboriginal and Torres Strait Islander practitioners' communities. Many people hold the view that choosing to undertake child protection work is indicative of 'becoming assimilated into such a system and walking away from their family and community' (Green et al. 2013, p. 210). Some of the participants in this study similarly described receiving negative responses from their family and community for undertaking child protection work, which sometimes affected their family life and wellbeing. Some families reportedly struggled to understand why their family member would want to work for the government, given the history between Indigenous Australians and child protection authorities:

> I think because a lot of her family, them being part of the Stolen Generation—because of what happened with my great grandmother and the fact that they were forced to give up their culture, they sort of view it as a bit of a betrayal in some ways. So a lot of the extended family struggle to understand how I could stay in that role for as long as I did. (Alice)

> It is difficult for Indigenous workers because they've got dilemmas. They have the pull between family responsibility, staying with your cultural roots, your values there, or working within a very structured system where history has not been kind to Indigenous people. (Elvina)

Many participants discussed balancing the expectations of their community with the professional expectations of their role as child protection workers:

> Because there's that sense of community responsibility and community expectation for us and that everything that you do should be of benefit for the community as a whole. (Alice)

The participants sometimes found community-based pressures challenging; however, no one described their key position within their communities as unwanted. Although

described as frustrating and difficult to juggle, the examples shared by the participants identified how their role in their communities was shaped by their work and was shared with a sense of pride. Being the 'go-to person' within a community because of their child protection knowledge was described by some participants as a positive aspect of their work:

> One of the positives is that I can give my family information about how investigation works if something happens. (Alice)

### 3.3. Confidentiality and Family

Many participants shared examples of pressures that can arise from within their family group as a result of their work in the child protection sector. Some participants shared the difficulties they have experienced when members of their family find themselves being investigated by statutory child protection services. The participants shared how challenging it can be to manage regular requests to obtain confidential information, and the precarious position that places them in:

> He called me up straight away as soon as they [child protection services] left and told me what had happened and asked me if I could look in the system. When I told him, 'I can't do that. This is confidential information. It would be me basically ignoring my contract, ignoring all of the things that I had signed', and then he put a lot of pressure on me. 'Well, what's worth more, your contract or your relationship with your family?' That's not fair ... Again, because of gender and age and generational issues, I didn't feel comfortable saying to him, 'Well, maybe you shouldn't have been involved in DV [domestic violence] with your partner', although that's what I was thinking. You know, 'Don't hit your wife. That's not okay either. We wouldn't be in this predicament if that hadn't happened'. But of course, I couldn't say that to him. (Alice)

Some participants described a refusal to disclose confidential information leading to being ostracised by their family for a time. Some of the ostracising behaviour experienced by participants included being whispered or gossiped about, being excluded from family events and being verbally and physically threatened:

> Because you work there, so you should stop them or you should have warned me. We get that a lot. (Sarah)

All participants who shared experiences of managing family and community requests for confidential information used the strategy of reiterating that they were not in a position to share that information, and directing people to the appropriate avenues:

> You just kind of have to wear your privacy and confidentiality on your sleeve and it's like, 'Can't talk about that, can't talk about that, sorry', but give them opportunities and avenues of where they may be able to seek that information. Or if it's straight out, 'It doesn't matter where you go, brother, you're not going to get that information. That's that person's information'. So it's hard, but you know—it's dinners, it's family events, birthdays. (David)

> Sometimes people approach me in shopping centres, but they mostly know me from when I worked at [name of workplace] and they know I can't say anything. (Veronica)

Although the participants described working in their own community as difficult at times, they shared a great sense of pride that they were in a position to help and give back to their communities. The participants also described a deep sense of comradery with other Indigenous child protection practitioners because they shared a unique and important role within their communities.

*3.4. Community Expectations, Visibility and Availability*

The participants confirmed that there is often an expectation within the Aboriginal and Torres Strait Islander community that workers will be available in an informal capacity outside of their paid work hours to manage any child protection–related matters that may arise. This could include questions about processes, such as how to make a child protection notification and where a person can seek parenting support. In addition, the participants needed to manage community expectations, such as requests for confidential information regarding open investigations as mentioned previously, and the expectation to support children who arrived unexpectedly at their houses needing somewhere safe to stay:

> It never leaves you, it finds you everywhere you go. (Isabella)

> People have come to my house to notify concerns and I listened and gave advice. (Veronica)

> I got pulled up putting petrol in my car one day by a family who just recognised me and, yeah, just came and had a chat with me while I was putting petrol in my car. So, you can't run away from it . . . you kind of forget it because it just happens. It's common. It happens to all of us. (David)

Indigenous writers Calma and Priday (2011) and Menzies and Gilbert (2013) argued that balancing roles can be challenging for Aboriginal and Torres Strait Islander practitioners, particularly for those who work in their own community. The participants in this study also identified challenges and difficulties associated with these high expectations:

> She got really burnt out [another more experienced Indigenous child protection practitioner]. There was a lot of pressure on her from her community. There was a lot of pressure from the office and she got very ill towards the end. (Alice)

> I have to live out there. I have to go out and remove children from families that probably have relationships with the extended family that I know. You know what I mean? I've got to engage with these people. I go to community functions. I go to NAIDOC [National Aboriginal and Islander Day Observance Committee]. I go to openings. I go to funerals. Are you going to engage with these people? There's no acknowledgement how difficult that is when you've just gone and snatched their children. Or they blame you. (Sarah)

However, the participants shared both benefits and challenges that arise from working in the same community in which they lived. The participants discussed that, regardless of the size of the place one lives, the Indigenous community is very small in comparison to the non-Indigenous community. Additionally, the participants described that knowing family ties and cultural links is an integral part of Australian Indigenous identity, rendering them even more visible within their communities. Some participants described feeling a consistent pressure to ensure that their work benefited their community, rather than contributing to further disadvantage and trauma:

> It can be really, really beneficial and then it can be not so beneficial . . . you want to help your community, but at the same time, being local, [name of town] is a small place, but when you're Indigenous, it's even smaller. It is difficult . . . You want to help, but it can be hard. I think because the Torres Strait Islander community is very small and everyone knows your family, so I think once they make the connection of who you belong to, it can be really difficult. (Grace)

> I was constantly asking questions because I needed to be at peace with myself and those decisions and go home and try and function with my family and my community. (Sarah)

The participants described this constant visibility to be challenging at times. Moreover, they further stated that the increased visibility because of their work in child protection could also extend to other members of their family, including their children:

Like I said before, a member of the community that you work in, there's no anonymity. People know where you live or, if they don't, they're going to find out through family. I used to have people going past my house and swearing at me. I used to have to shop not at the shopping centre down the road, but one three hours over. It's not just clients. It's carers. Everyone wants a piece of you constantly . . . it's not 9.00 am to 5.00 pm. (Mary)

*3.5. Organisational Expectations and a Lack of Cultural Safety*

Many participants shared experiences that related specifically to their Indigeneity and their work in the child protection field. The participants described feeling a sense of expectation and pressure from their agency because of their Indigeneity. For example, many participants described their agency's expectation that they would know all the details about every Aboriginal and Torres Strait Islander issue:

I guess the other issue that I always found was if you become recognised as being Indigenous, you're supposed to know everything. I'm from [name of town] and, until I moved here, I'd never seen a Torres Strait Islander person in my life. That's a very different culture to Aboriginal. (Mary)

I think as a Black worker in a White organisation, it can be really, like, you get pressure. (Grace)

Happens all the time. Walk into a room, you're the only one there. Walk into a room and Welcome to Country has to happen—it doesn't matter where you could be from . . . they ask you to do it. (Isabella)

Some participants described an internal struggle or even fear of not meeting the expectations of their workplace because of their exposure to many of the same issues with which clients of the child protection system struggle:

The big thing that I've told a lot of professionals and colleagues too is there seems to be this perception . . . that because you're a professional . . . you are immune to the transgenerational trauma as an Aboriginal and Torres Strait Islander person . . . we're members of this Aboriginal and Torres Strait Islander community and suffer the same trauma that a lot of our clients do. There's just an expectation that you're immune . . . the trauma doesn't affect you. You're different. (Sarah)

I had fears of me becoming a [statutory child protection worker] because of my personal history. (Rosalyn)

Some participants discussed their constant anxiety of not being good enough or being unfairly judged by non-Indigenous colleagues:

I operate with a certain level of nerves all the time because I'm a Blackfella in a Whitefella's game. (Isabella)

I'm a human being. I had issues in my life. There shouldn't be any shame attached to that. (Sarah)

Other participants described altering the way they spoke to appear more professional in the eyes of their non-Indigenous colleagues:

Language for us is—so we talk about broken English and proper English. We don't speak proper English at home. So we constantly straddle two worlds. And so I struggle sometimes even now where—it's really horrible because you have these moments where you're like, I'm dumb, I don't get this stuff. And it's because—we just—and sometimes I actually cannot find the words to describe it. And I try to. (Isabella)

Niemann (2003) argued that minority groups are 'continually aware of putting their best foot forward so as to not negatively affect perceptions of other members of their demographic groups . . . they seem to be in a situation of constantly examining and second guessing their behaviour' (p. 4). Although Neimann is an American researcher

researching minority groups in the United States, the experiences of participants in the current study were remarkably similar to those interviewed in Neimann's study. The participants described consciously limiting the information they shared about their family and/or upbringing to appear more 'mainstream':

> My Dad used to have very, very deliberate and conscious discussions with us at a very young age about how to behave with White people and how to behave with Black people and why you do it—so very open, he prepared us for it. (Isabella)

Some participants said that they do not share information about their family and community where dysfunction is present with colleagues in their workplaces. The participants explained that they did not share these experiences for fear of judgement, having their professional skills questioned, and perpetuating existing myths about Indigenous people. Some participants described this phenomenon as having to be the 'model Black':

> I have to admit that I would worry that she would question my ability to do my job. And so I will not put myself at risk like that. So she will know when she [line manager] needs to know. And then she will know enough of what she needs to know to support me to maybe take a day off. (Isabella)

> It can't have touched your life at all because you're a good Black. You've got a job. (Mary)

## 4. Support and Supervision

### 4.1. Internal Supervision

Participants were asked specifically for their experience of supervision in their workplaces. All participants defined internal supervision to be supervision that occurs in their workplace, usually provided by their line supervisor. The participants' experiences of internal supervision varied depending on whether their agency provided statutory or non-statutory child protection services. The participants who spoke of their experience of internal supervision while employed by non-government, non-statutory organisations reported that they had good access to supervision that was supportive and useful:

> It is fantastic . . . community sector are much better at it, much better at supporting workers. (Matilda)

> It's very good. (Missy)

> It's been really good. (Grace)

However, the participants with statutory child protection work experience largely reported poor access to internal supervision, which they described to be administrative, task focused, compliance based, inconsistent and often not meeting their professional or practice needs:

> It is very, very tasky. (Isabella)

> My idea of supervision coming from social work was very different to what I found when I came to [government department] because the idea of supervision in [government department] was, 'Let's look at your board. Let's look at what cases you've got. Let's talk through that', and that usually takes an hour. 'Have you got any leave coming up?' 'No.' 'Alright, fine. You may leave now'. Whereas social work supervision is more critical reflection: How can I improve my practice? How can I link my theoretical understanding of social work practice to what I'm actually doing on the ground? How can I engage with some self-care practices? That was my idea of supervision and so at first I was a little disappointed by the supervision that was available in the department. The only way I could solve that was by seeking external supervision with the social worker outside of the department. (Alice)

Many of the participants stated that the supervision available to them in their statutory child protection workplace did not tend to focus on their wellbeing or self-care needs:

It certainly wasn't about my wellbeing and it was generally me organising it. But definitely not about me, that's why I burnt out, because I had no support. (Matilda)

Supervision was more about getting the job done, paperwork-based type thing, compared to worrying about how I felt or what I was going through at the time. It was very bang, bang, bang, get the job done, bang, bang, bang, get the paperwork done. 'You should have this, this and this done by now. Why haven't you?' (Rosalyn)

Many participants described a lack of emphasis from their supervisors regarding the importance of supervision. The participants often experienced supervision sessions being postponed or cancelled to accommodate more 'important' matters:

I think the biggest thing is inconsistency and that's why I ended up finding the issue is, yeah, we have one month and then we reschedule for the next one because we were either both busy or our schedules didn't allow for a supervision session because something pops up. So you've got to go and deal with that. So it needs to be set in stone, it needs to be regular, it needs to be consistent. (David)

Some participants stated that regular formal supervision was never scheduled in their statutory workplace:

I could probably count on my two hands the number of times I had supervision in that [number] years. (Matilda)

I wasn't a statutory officer. I didn't require supervision, but even the statutory officers didn't get supervision. (Sarah)

### 4.2. External Supervision

Almost all participants stated their preference for external supervision over the internal models of supervision that would usually be delivered by their line supervisors. The participants had a congruent definition of what they believed external supervision to be—that is, private sessions with an external experienced practitioner during which work and wellbeing matters can be discussed. All participants stated that they had either participated in external supervision while working in the child protection field or knew of colleagues who had. Some participants spoke specifically about the difficulty of being constantly exposed to the level of trauma they encountered as part of their work in child protection, and shared that, for them, external supervision was an essential part of managing this trauma:

I think it needs to happen everywhere, and particularly in somewhere like child protection because I think that—the issue is, is that people kind of lose themselves in that organisation because they just become so entrenched in the trauma and the, you know, the heartache that goes on. Their hearts either become super hard, or they break. (Matilda)

You're able to go and get things off your chest. The weight off your shoulder. For your own mental wellbeing, because what we take on every day is huge. To be able to talk to people and get stuff off your chest on hard cases is important. (Missy)

I think that outside supervision is good just to have that, you know, if stuff's building up and you're not able to talk to anybody, then you've got somebody that you can actually go out and talk to and it's somebody that you've chosen yourself. (Lisa)

Many participants shared that external supervision offered them a place of safety where they could discuss matters—both personal and work related—without fear. Participants used words such as 'freedom', 'safe' and 'private' to describe their experiences with external supervision. The participants expressed a reluctance to express their needs fully

during internal supervision primarily because of fear of being judged unfairly by their supervisor or being viewed as incompetent:

> It just gave me the freedom, I suppose, to feel like, okay, whoever I'm talking to, it's definitely not—there's no connection to the department in any way. (Alice)

> Yes. Definitely. I can tell her anything and everything and she listens and advises me. But yeah, she doesn't have that connection back here. Because I feel you just don't—when you work in a place where you're not fully trusting of everyone within that organisation, it can be a concern if you want to go yarn to someone about something. So the fact that you can go and talk to someone totally outside of the org[anisation] is good. Very good. (Missy)

The participants reported experiencing substantial benefits from external supervision, particularly in relation to worker wellbeing:

> I don't think I would have lasted with [government department] or in child protection. I probably would have started to be one of those jaded people that thought child protection was a horrible place. (Alice)

Some participants viewed internal supervision in a statutory child protection context to be about tasks and case-related discussion, whereas external supervision was about them and their support and development needs:

> I think it's good to have supervision with your team leader. I think that's required, I think that's needed. I think cases need to be discussed. But on the other hand, they need to have that emotional debriefing about how it's impacting on them along the way. (Rosalyn)

> I think because I was getting the case discussion supervision with my team leader, I felt like my time with my external supervisor could be more focused on practice development and self-care. (Alice)

While the participants only reported positive experiences with external supervision, they did discuss a number of issues related to access. Two main access issues emerged from analysis of the participants' narratives: cost and time. The cost of contracting an external supervisor without the financial support of the workplace, at least in part, was described as a factor that would restrict access to external supervision for many practitioners:

> You know, $150 or $160 a month for a session is sometimes out of our reach . . . they give you the time off, but you have to actually pay for it. (Elvina)

> I think if the department was to say, 'We will give you this money for some external supervision', then I think more people would take it up. I would like to advocate for [government department] to fund that for their staff because I think it puts a lot of pressure on if you're expected to fund it yourself. (Alice)

Others described the issue of time as a critical factor pertaining to accessing external supervision. The participants stated that, because of their heavy workloads, even if the workplace approved time for external supervision, they felt they would be unable to access it. Many stated that they would need to access external supervision during worktime because of heavy family commitments after hours:

> I'm happy to cover the cost, but I need to do it during work time. So I've written that into my plan. (Isabella)

> I mean, I had asked for it at [government department], but, yeah, it was 'pay for it yourself, do it in your own time'. (Matilda)

### 4.3. The Supervisor

Emerging from the participants' narratives were a number of similar attributes held by 'good' supervisors. The participants all described trust as critical to their ability to freely participate in and benefit from external supervision:

And the other thing is, is I need to keep that completely confidential ... completely external and—and private. (Isabella)

But every time I see her, I do feel good. I do feel like she listens and she gives me good advice. (Missy)

It has to be a regular thing and it has to be in a cone of silence, so nothing is reported back, and that trust in that supervision should be there. (Elvina)

I asked participants specifically for their views on whether supervision for Indigenous staff should be provided by an Indigenous supervisor. There was no consensus among the views of the participants in relation to the cultural background of the supervisor. Some participants stated they did not have a preference for an Indigenous or non-Indigenous supervisor:

Well, it's like my supervisor coming into that system supervising me, whether it's black or white or whatever origin. The supervisor just needs to have those skills. (Elvina)

Other participants stated that they would prefer an Indigenous supervisor; however, if one was unavailable, then a non-Indigenous supervisor would be suitable:

I had a couple of people at work that I could talk to—that was good enough for me. But I think if an Indigenous person had an Indigenous person to speak to, they would have better understanding of the historical stuff and be able to identify and relate to some of the issues that that person was going through at the time. (Rosalyn)

One participant stated that having access to an Indigenous supervisor was a non-negotiable for them:

There's just something about our shared experience and our shared inherent histories that connects us more than what it would to a white person. So, like, if you talk about Stolen Generation, she's going to have her story, I'll have my story. That's what connects us. Whereas it's not going to be the case for white people. (Isabella)

The varied views of the participants indicated that Indigenous child protection practitioners do not have homogenous supervision and support needs. Berlin (2002) argued that:

Classifying people on the basis of group membership only gives us the illusion that we are being culturally sensitive, when, in fact, we are failing to look beyond easy characterisations for the particular and specific ways that this person is understanding, feeling and acting. (p. 144)

Moreover, participant Matilda stated:

We're all individuals. You know you can't say all Aboriginal people are the same, and all Torres Strait Islanders are the same. We all come from different backgrounds, you know, it's not just about the culture itself, it's about all of those other cultural things, like, you know, what family means to us—we are part of a group, another sort of group. To me, I think that it's more about—I think it's more about having information and having support available to all staff when it comes to cultural matters. (Matilda)

There was consensus among participants that the provider of supervision must have child protection work experience. All participants expressed that a child protection work history was an essential element:

You do need a likeminded person to debrief with. (Elvina)

Yeah, just so that they can understand what it is that you're actually talking about and why you're perhaps feeling that way. (David)

I have two people that I go between because they give me different needs and both of them, particularly, have worked in the Indigenous space, as in with

children and families, workers, organisations. So those two are the only ones that I'll go to. I won't go to anybody else. (Isabella)

Some participants expressed a sense of comfort and freedom in their sessions when speaking with a supervisor who had also worked in the child protection field:

I tend to use humour as a defence mechanism, but because of my work in child protection, I have a very dark sense of humour. So I would tell jokes during our sessions and if I had told that joke to somebody who didn't have that background, I think it would have been more uncomfortable, whereas for her, she got the joke, she got the fact that I needed to use my sense of humour in that way. Yeah, so that was also good. That was very liberating. (Alice)

Because she's got a strong child protection background, she's able to give me some good advice as well. If it was just a psychologist that was general and didn't know the child protection stuff, I think I'd just be trying to explain myself a bit more. Whereas I can just straight off the bat yarn to her about a case and she knows exactly the processes and exactly what should have happened or should be happening and will advise me on what to do. (Missy)

All participants categorically agreed that external supervision was their preferred model of supervision, particularly in relation to managing matters pertaining to wellbeing.

The literature has noted that practitioners' feeling of safety within the supervisory relationship is a critical element in quality supervision and staff support (Davys and Beddoe 2010; Pack 2015). The experience of the current study participants is consistent with the work of Pack (2015) and Davys and Beddoe (2010).

### 4.4. Other Supervision Models

Some participants also discussed other models of supervision. Peer and cultural supervision models were communicated by these participants as having been useful to them while working in the child protection field. I will present and discuss both models in turn.

Peer supervision was defined by the participants to be an informal meeting of Indigenous child protection practitioners, either in pairs or smaller groups, where unguarded conversation could be held in a safe and peer-supported environment:

It's like these three Black women just come together and so two of us are working in non-Indigenous organisations and it's really lovely because—and this is the stuff I really struggle with and the people don't actually realise—is that you constantly straddle two worlds. We find support within one another. (Isabella)

Whenever we came back from the difficult one or anything like that, the three of us most times at the end of the day would sit down and have a little debrief and a chat about what it was we dealt with that day and was there anything heavy. (David)

There was another lady who worked in the field and me and her used to just go into one of the interview rooms and have a cry and a hug and talk it through. We would very much relate to the issues that were very raw for us within the office. (Rosalyn)

There's no cultural supervision. There's not even an acknowledgement of the need for it. Realistically, I suppose that's why then, within the workplace, you connect to other Aboriginal and Torres Strait Islander workers within the workplace. (Mary)

The experiences shared above by Isabella, David, Rosalyn and Mary all acknowledge that a shared sense of experience gave them a feeling of trust, safety and subsequently support.

Cultural supervision was defined by those participants who had experience with this form of supervision to be supervision that involves an Indigenous supervisor, where per-

sonal, cultural and practice matters can be discussed. Not all participants had experienced receiving or providing cultural supervision. Bessarab (2013) defined Aboriginal cultural supervision to be 'another form of professional supervision where the worker is supported so that they understand and apply a cultural analysis to their practice, aided by the supervisor, thus building their cultural competence and knowledge' (p. 75). However, Bessarab (2013) acknowledged that there is a dearth of literature related to cultural supervision and its application and effectiveness in an Australian context. I asked the participants their views of cultural supervision as a support model for Aboriginal and Torres Strait Islander child protection practitioners:

> I think it should be made available. I think you need to support Indigenous staff. I think that's really, really important to support Indigenous staff to just stay. (Lisa)

> It would be up to them for when they need it, but have that freedom to do it when they need to do it. (Rosalyn)

Of the two participants who had experience with cultural supervision, both reported the experience to be positive and beneficial:

> With the cultural supervisor, it's made a big difference. I'm feeling a lot more comfortable, in sharing personal too, I suppose . . . I think because our cultural supervisor has a better understanding of some of my personal struggles too, especially being a Blackfella, and then they can also advocate—well, you know, talk to our team leaders about some stuff too where I may not feel comfortable . . . it's been really good having a cultural supervisor here. (Grace)

> I think it's a very important position in the organisation because I think that—I think it just gives Indigenous staff a point to come and talk to about anything and everything. (Lisa)

Both these participants also noted that they would like to see cultural supervision provided to non-Indigenous staff, as well as Indigenous staff. They believed this would give non-Indigenous practitioners access to an experienced Indigenous practitioner for ongoing professional learning and development to strengthen their cultural practice competency. This position is supported in the literature by Bessarab (2013), who argued that both Aboriginal and non-Aboriginal practitioners 'often require appropriate Aboriginal cultural supervision that can assist them to address and understand the cultural and political issues emerging in their work' (p. 75).

## 5. Discussion

The findings from this study indicate that Aboriginal and Torres Strait Islander people who undertake child protection work have a unique position in their families and communities as well as in their profession. Many of the participants viewed their role in child protection as giving back to their people and communities for their own privileged upbringing. Participants framed 'privilege' as being raised securely within their culture. Participants shared a deep desire to support Indigenous children, families and communities where there are concerns, to remedy those concerns and divert them away from the child protection system.

This study highlights the multifaceted complexities experienced by Indigenous people who practice in the field of child protection. As an example, while some of the family responses to practitioners entering the child protection field were positive, many participants spoke at length about the negative responses they experienced from their family, community and other Indigenous people. Australia's history of systematically eroding the cultural, social and economic fabric of Aboriginal and Torres Strait Islander communities has created ingrained fear and mistrust of authorities that continues into a contemporary context (Harms et al. 2011; Menzies and Gilbert 2013; Walter et al. 2013). Having a family member move into working in the same systems that communities are deeply suspicious

of, can be extremely difficult for practitioners and their families and communities to navigate. An inherent power imbalance exists between those who have been oppressed and those seen to represent the oppressors, such as child protective authorities and other government representatives (Briskman 2007; Menzies and Gilbert 2013). In this context, the negative reactions of the participants' families towards the participants' work for child protective authorities are not unexpected. Although not unexpected, this feature of the lived experience of Indigenous people undertaking child protection work does need to be considered by non-Indigenous supervisors and managers in relation to the kind of support and supervision available to practitioners.

### 6. Implications for Practice and Recommendations

Participants in this study were clear in conveying that Indigenous practitioners are not a homogenous group with homogenous professional support needs. Given the strong consensus from practitioners that external supervision is their preferred method of support, consideration needs to be given to how workplaces can provide this level of support and supervision in order to meet the needs of Indigenous practitioners.

As previously outlined, the numbers of Indigenous children coming to the attention of child protection authorities in Australia is increasing. While additional recruitment of Indigenous practitioners as a strategy to reduce disproportionate representation has merit, there is a lack of research and practice knowledge about how best to support Indigenous people in the child protection workplace. What constitutes a culturally safe workplace inclusive of responsive support and supervision models, requires further investigation led by Indigenous practitioners themselves. The participants in this study have facilitated the start of a conversation about their unique position working in child protection and the need for support and supervision that recognises, acknowledges and plans for the multifaceted experiences of Indigenous practitioners. It is our recommendation that further exploration of the lived experiences of Indigenous practitioners be undertaken both in Australia and internationally with a particular focus on those countries where state-sanctioned child removal featured as part of colonisation. These explorations would inform what responsive support and supervision models would best support the unique needs of Indigenous people undertaking child protection work as defined by Indigenous practitioners themselves.

**Author Contributions:** Conceptualization, F.O.; methodology, F.O. and K.M.; software, F.O.; validation, F.O.; formal analysis F.O.; investigation, F.O.; resources, F.O. and K.M.; data curation, F.O.; writing—original draft preparation and review and editing, F.O.; supervision, K.M.; project administration, F.O.; funding acquisition, F.O. All authors have read and agreed to the published version of the manuscript.

**Funding:** This research was funded by an Australian Government Research Training Program Scholarship.

**Institutional Review Board Statement:** This study received ethical approval from James Cook University Human Research Ethics Committee (approval number: H6266).

**Informed Consent Statement:** Informed consent was obtained from all subjects involved in the study.

**Conflicts of Interest:** The authors declare no conflict of interest.

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
