# Peer review of "Working for the Welfare: Support and Supervision Needs of Indigenous Australian Child Protection Practitioners"

_socsci, doi:10.3390/socsci10080277_

Round 1

Reviewer 1 Report

Excellent paper; methodology was ideal for this study.  A few questions: line cite the anthropological scholarship that contributed to the beliefs that aboriginal people were incapable of "evolving.  "  Define disproportionate. That term has been used in the US to argue that blacks are disproportionately confined in prisons. Those who argue that compare the % of blacks in the general US population v. US prisons.  But Simpson's paradox shows that the percent of blacks in each state prison system is well below the % of blacks in each state.  All qualitative researchers do not embed themselves in the lives of the folks they study. But when we get very close to the folks we study we must come to terms with how our personal beliefs bias our interpretations. In a short paragraph explain how you, the primary author, kept control of how you felts seeing the awful mistreatment of aboriginal people. Some readers might dismiss your excellent work arguing that you are biased by your close engagement with aboriginal people. Language preservation was omitted. I mention this because I've studied American Indian languages with only a few native speakers.  

Reviewer 2 Report

Wow! What a fantastic piece of work! I loved reading this article, learned so much from it, and was so impressed the authors' clarity of analysis and presentation of findings. This is very important work and everyone working in child protection--in Australia and beyond--should read it.

In particular, I found the weaving of participant voices throughout, to be of incredible value. I highly recommend this piece for publication and believe that it makes an invaluable contribution to the field of child protection—in Australia for sure, but also internationally as many of the dynamics documented in the study’s findings are in play and relevant in other contexts. Congratulations on a great piece of work and thank you for the opportunity to review this article.

More specifically:

  • I found the context setting early in the document amazingly clear and well put together—and helpful to interpreting the findings that follows. 
  • The analysis is superb. The points made do such a nice job of highlighting and driving home the complexities that these practitioners face. In particular I found sections 3.1-3.5 incredibly insightful and interesting.

Also, some more specific comments here that the authors might consider—some of which are very minor:

  • The first paragraph in the introduction is a little dense and hard to make it through. I wonder if this might be addressed by simply breaking up the one paragraph into several shorter ones based on common ideas—this might help a bit with readability.
  • On the first page, lines 37-41: it would be helpful to explain the rationale and make it explicit as to why recruiting more indigenous partitioners helps to decrease over-representation. The authors take this connection for granted but I feel like it is important to the piece and that it would be helpful to lay it out and be explicit about the connection, and provide any relevant citations that back up the connection (i.e. that increasing the number of Indigenous practitioners decreases/addresses patterns of over-representation of those same groups in the system).
  • There are a few typos that could be fixed:
    1. Sentence on line 147 is missing a word
    2. Something with the sentence on line 161 isn’t quite right
    3. Sentence on 193 isn’t quite right
    4. “understated” on should be “overstated” I believe
  • It would be good to have a bit more detail in the data collection and analysis section. Are these methods that have been used elsewhere (citations)? What’s the theory that underpins these methods (i.e. why were they used over other interview/analysis methods?).
  • My most important comment is that I think the discussion could be expanded slightly to include more interpretation and implications of the findings for practice and more specificity around the ways that they suggest Indigenous practitioners need to be supported.
